# Immediate Loading of Implants Placed by Guided Surgery in Geriatric Edentulous Mandible Patients

**DOI:** 10.3390/ijerph18084125

**Published:** 2021-04-13

**Authors:** Eugenio Velasco-Ortega, Alvaro Jiménez-Guerra, Ivan Ortiz-Garcia, Jesús Moreno-Muñoz, Enrique Núñez-Márquez, Daniel Cabanillas-Balsera, José López-López, Loreto Monsalve-Guil

**Affiliations:** 1Comprehensive Dentistry for Adults and Gerodontology, Faculty of Dentistry, University of Seville, 41009 Sevilla, Spain; evelasco@us.es (E.V.-O.); alopajanosas@hotmail.com (A.J.-G.); ivanortizgarcia1000@hotmail.com (I.O.-G.); je5us@hotmail.com (J.M.-M.); enrique_aracena@hotmail.com (E.N.-M.); danielcaba@gamil.com (D.C.-B.); lomonsalve@hotmail.es (L.M.-G.); 2Faculty of Dentistry, Service of the Medical-Surgical Area of Dentistry Hospital, University of Barcelona, Hospitalet de LLobregat, 080997 Barcelona, Spain

**Keywords:** guided implant surgery, immediate loading, edentulous mandible

## Abstract

The aim of this study was to show the clinical outcomes of the immediate loading of implants inserted by guided surgery in edentulous mandible patients. Edentulous mandible patients were diagnosed with oral examination, cone beam computerized tomography and diagnostic casts for intermaxillary relations and treated with 8–10 implants for rehabilitation with guided surgery and immediate loading. After flapless surgery, implants were loaded with an immediate acrylic temporary prosthesis. After a period of six months, a ceramic definitive full-arch prosthesis was placed. A total of 22 patients (12 females and 10 males) were treated with 198 implants. Eleven patients (50%) had a previous history of periodontitis. Six patients (27.3%) were smokers. The follow-up was 84.2 ± 4.9 months. Clinical outcomes showed a global success rate of 97.5% of implants. Five implants were lost during the healing phase with provisional prosthesis. Twenty-two fixed full-arch rehabilitations were placed in the patients over the 193 remaining implants. Mean marginal bone loss was 1.44 mm ± 0.45 mm. Six patients (27.3%) showed some kind of mechanical prosthodontic complication. Eighteen (9.3%) of the 193 remaining implants were associated with peri-implantitis. The antecedents of peri-implantitis are critical elements for the survival of the implants. The loss of implants was significant in patients who smoked up to 10 cigarettes, compared to non-smokers. Peri-implantitis is one of the key elements in the long-term follow-up of implants and it was more manifest in smoking patients, and in those with a history of peri-implantitis. Marginal bone loss was more significant in smokers. Full-arch rehabilitation is presented as a predictable alternative with minor fatigue problems that are easily solvable.

## 1. Introduction

Guided dental implant surgery is increasing in popularity today, particularly due to the advances in, and increased usage of, Cone Beam Computed Tomography (CBCT) and the development of dental implant treatment planning software that allows for a three-dimensional assessment of the implant site. Preoperative planning of the implant position, as part of a comprehensive prosthetic and surgical approach, is becoming increasingly important regarding function and esthetics [1,2,3].

The CBCT provides a noninvasive method to describe maxillofacial structures and assess bone volume and density of alveolar ridges. The use of CBCT-based implant planning succeeds in fixed surgical procedures with a high level of precision in the edentulous maxilla and mandible. The introduction of specific software for guided implant dentistry can improve the virtual planning of flapless surgery and the outcomes of dental implants placed in edentulous alveolar ridges by template guided surgery [4,5,6].

Dental implant planning is currently based on complex diagnostic imaging. The exact positioning of the fixtures is one of the most important goals of the surgical phase in implant dentistry. Today, implant surgery is based on improved diagnostic technologies that give the clinician more accurate information on the maxillofacial anatomy of patients, allowing the surgeon to dynamically interact with a 3D digital reconstruction and plan to evaluate the surgical approach [1,6]. This virtual treatment plan can be transferred to clinical practice through the use of surgical guides, which allow implant insertion in the ideal predetermined position, virtually. One main outcome of computer-assisted surgery is the possibility of inserting implants in a more accurate manner in limited bony volumes, using a precise guide, and these modern techniques allow less invasive approaches, such as flapless surgery, and simplify the prosthetic procedures involved with immediate loading protocol surgery [7,8,9,10].

Many studies have shown the clinical effectiveness of this implant treatment technique [11,12,13,14,15,16]. Based on the scientific literature, computer-assisted implant surgery is a safe, less morbid, and efficient alternative of implant dentistry because CBCT planning and flapless techniques are beneficial to improve the clinical outcomes of patients [11,12]. In fact, edentulous patients can be treated with several implants for rehabilitation with guided surgery and immediate loading [8,13,14]. The immediate functional loading of implant-supported fixed full-arch prostheses can now represent a predictable solution for the rehabilitation of edentulous patients [15,16]. In many cases, the immediate loading protocol maximizes the success of the guided surgery techniques with many benefits, such as shortened time and maximum patient comfort [11,12,13,14].

The aim of this study was to investigate the clinical results of guided surgery of implants and immediate loading with fixed full-arch prostheses in the treatment of geriatric edentulous mandible patients.

## 2. Materials and Methods

This clinical study included geriatric edentulous mandible patients presenting for treatment in the clinic of Master of Implant Dentistry at the School of Dentistry of Seville, Spain, from January 2011 to December 2015. The study was conducted according to the principles outlined in the Declaration of Helsinki on clinical research involving humans. The ethical committee of the University of Seville approved the study, and informed written consent for implant placement was obtained in all patients.

The study population consisted of 22 patients (treated consecutively), 12 females and 10 males, ranging in age from 62 to 77 years (mean age 65.4). The inclusion criterion was the need for mandibular full implant supported rehabilitation. The exclusion criteria were the presence of chronic systemic disease, smoking ≥10 cigarettes/day, bruxism, uncontrolled diabetes or periodontal disease, coagulation disorders, and alcohol or drug abuse. Thus, the patients included in this study are ASA I-II patients, without decompensated systemic diseases or medication that may interfere with the osseointegration of the implants. Treatment planning included oral examination, cone beam computerized tomography, diagnostic casts for intermaxillary relations, and clinical photographs. Patients were informed of all possible implant treatments and accepted the immediate implant-supported prostheses by guided surgery.

Prior to surgery, the patients received preventive antibiotic therapy (500 mg amoxicillin and 125 mg clavulanic acid 1 h before surgery); they also continued to take the antibiotic, postoperatively (3 capsules daily for 7 days). All patients were treated under local anesthesia using articaine with adrenaline. After surgery, a chlorhexidine mouthwash was prescribed for twice daily use for 30 days. Ibuprofen (600 mg, 4 times daily) was prescribed for 7 days.

All participants underwent cone beam computer tomography (Picasso Master 3D^®^, Vatech, Gyeonggi-do, Korea) with a scan prosthesis and occlusal index positioned in the mouth. The implants were planned in 3D software (Galimplant 3D ^®^, Galimplant ^®^, Sarria, Spain) in the optimal position, considering both the alveolar process and the prosthetic demands. Figure 1a,b.

A flapless surgical approach was chosen with the help of an image-guided template. After the digital planning, the surgical template was placed in the mouth. In all patients, the template was secured to the underlying bone with two screws in the vestibular plates to avoid movement during the surgery. The guided surgery started with the preparation of all implant sites, using drills of incremental diameter, and ended with the placement of all planned implants, through the guide. Surgimplant ^®^ screw implants (Galimplant^®^, Sarria, Spain) with sandblasted and acid-etched surfaces and external connection were used for all implant placements. Insertion torque and resonance frequency analysis were used as methods for measuring implant stability after placement. Insertion torque was measured before the removal of the surgical guide. Since all implants were placed using the implant motor, a standard insertion torque of ≥35 Ncm was set at placement [8,14]. Finally, resonance frequency analysis was used (Penguin RFA^®^, Clokner, Barcelona, Spain) to confirm the stability of each implant, immediately after removal of the surgical guide once the implants had been placed. The stability of the fixture was considered acceptable with an implant stability quotient that ranged from 55 to 85 [16]. Figure 2a–c.

After the surgical procedure, all patients immediately received abutments and a temporary prosthetic restoration. Immediate loading was performed when an insertion torque of ≥35 Ncm and ≥55 ISQ value (resonance frequency analysis). Acrylic-resin cement was used for mandible full-arch temporary restorations. Six months after implant placement, temporary restorations were removed. Impressions were made with addition silicone material using open individual trays. Definitive ceramo-metallic full-arch restorations were manufactured and placed onto the osseointegrated implants. Figure 3a,b.

The criteria used for the assessment of survival were implant stability and the absence of radiolucency around the implants, mucosal suppuration, and pain. Follow-up visits were scheduled at 3 and 6 months after implant placement and at 1, 2, 3, 4, 5, 6, and 7 years post guided surgery. In these revisions, the patients were subjected to cleaning and clinical and radiologic revisions of the prosthesis and implants. Marginal bone loss was evaluated based on digital periapical radiographs taken perpendicular to the long axis of the implants, comparing the difference between the 1-year follow-up radiography and the 7-year follow-up radiography. The analyzed records included patient information (gender, age, dental health, systemic diseases, and smoking habit), details about the placed implants (type, number, position, diameter, and length), and the prosthetic full rehabilitation (provisional acrylic prosthesis, fixed full arch restorations) including the dates of delivery. Further, the analyzed data included all information about any implant failure or biological and technical complication that occurred during the intervention, after the surgery and functional loading, and at each follow-up visit.

All available data from all examinations were included in the analyses using the SPSS (SPSS 11.5.0, SPSS, Chicago, IL, USA) package. Descriptive statistics were used to report the general results of the study. For all qualitative variables, values were expressed in absolute terms and in percentages (%) and were calculated using the chi-square test. For quantitative variables, the means, standard deviations (SD), medians, ranges, and 95% confidence intervals (CI) were calculated. The similarities in the groups were confirmed by analysis of variance (ANOVA). The Mann–Whitney U nonparametric test was used to compare differences between groups created, based on the different risk factors measured. A *p*-value < 0.05 was considered as statistically significant.

## 3. Results

In total, 198 implants were placed in 22 totally edentulous mandible patients, 12 females and 10 males. No significant statistical differences were found related to sex and age (chi-square test, *p* = 0.79856). A total of 11 patients (50%) had a previous history of periodontitis, 10 males and 1 female. These differences were statistically significant (chi-square test, *p* = 0.00542). Six patients (27.3%) were smokers, and 45.4% of patients with the previous history of periodontitis were also smokers (*n* = 5) (Table 1). All smoking patients were males. These differences were statistically significant (chi-square test, *p* = 0.03509).

Of the 198 implants placed in the mandible, 9 patients (40.9%) received 8 implants, 4 patients (18.2%) received 9 implants, and 9 patients (40.9%) received 10 implants. The average follow-up period was 84.2 ± 4.9 months (ranged: 76–84 months). A total of 87 implants (43.9%) had a diameter of 3.5 mm, and 111 (56.1%) implants had a diameter of 4 mm. In terms of length, 132 implants (66.7%) were 10 mm and 66 (33.3%) were 12 mm. A total of 5 implants (2.5%) in 5 patients (22.7%) were lost during the healing period before definitive loading with the ceramo-metallic prostheses due to a lack of osseointegration (Table 2). Loss of implants was more frequent in smoking patients (50%). These differences were statistically significant (chi-square test, *p* = 0.00104). The cumulative survival rate for all implants was 97.5%.

During the follow-up period, 18 (9.3%) of the 193 remaining implants in 10 patients (45.4%) were associated with peri-implantitis (Table 3). Peri-implantitis was more frequent in those patients with a previous history of periodontitis (63.6%) and was significantly more frequent in smoking patients (66.6%) (chi-square test, *p* = 0.0356).

The mean marginal bone loss was 1.44 mm (S.D. 0.45 mm), ranging from 1.2 to 2.1 mm during the 7-year follow-up evaluation. In patients with smoking habits, the marginal bone loss was 1.75 ± 0.33 for smoking patients and 1.34 ± 0.39 for non-smoking patients, with statistical differences (ANOVA; *p* = 0.00684).

Regarding the prostheses designed, a total of 22 fixed full-arch rehabilitations were placed in the patients over the 193 remaining implants after the healing period (six months). Six patients (27.3%) showed some kind of mechanical prosthodontic complications (Table 3). Additionally, 2 patients (9.1%) showed resin fracture of provisional prosthesis, and 4 patients (18.2) showed complications in definitive prosthesis (ceramic chipping, loss/fracture of the prosthetic screw).

## 4. Discussion

This study evaluated the clinical outcomes in planning and treatment by guided surgery of geriatric edentulous mandible patients with an implant-supported full-arch rehabilitation with immediate loading prostheses. A full rehabilitation of edentulous patients is always a challenge because optimal implant planning is strongly related with an accurate merge of the prosthetic and the radiographic data of bone availability. A correct diagnosis and accurate implant planning are key factors for success in full-arch rehabilitation [16,17]. The use of computer-based planning using CBCT allows the surgeon to reduce the risk of damaging nearby structures, especially in mandibular areas with limited residual bone [15,18,19,20]. In fact, the use of guided surgery is strongly recommended because this implant dentistry technique is more accurate than conventional implant surgery [21,22]. In these cases, CBCT planning can be used in edentulous mandible patients with anatomical limitations, such as the inferior alveolar nerve [19,20].

Computer-guided dental implant systems provide an elevated number of anatomical diagnostics, surgical approaches, and prosthetic evaluations for clinicians [1,2,3]. The use of this implant protocol allows for the insertion of implants with flapless surgery and the immediate delivering of the prosthesis. Immediate loading of implants placed in edentulous patients can be a reliable and predictable technique for full-arch rehabilitation [23,24]. Moreover, the overall satisfaction of patients with this computer-guided surgery and prosthetic rehabilitation is very high because the postoperative pain and discomfort is very low and improves the compliance in the functional and aesthetic outcomes of prosthodontic treatment [13,14].

The literature available suggests that computer-guided insertion of dental implants has an implant survival rate greater than or equal to those of conventional protocols [25,26,27]. Clinical implant outcomes related to computer-guided versus conventional surgery were investigated in the rehabilitation of edentulous patients treated with hybrid prostheses [28]. In this study, 45 patients were stratified, one group using computer guided insertion (149 implants) and another group using conventional insertion (111 implants), with a mean follow-up of 9.6 years. A significant difference was found between both groups, in terms of implant loss, with a lower incidence in the computer-guided group (3.3%) compared with the conventional group (19.8%). These results showed that computer-guided implant placement is a predictable alternative to the traditional approach for implant placement and immediate loading [28].

The results of several studies seem to confirm the evidence emerging from the literature in relation to computer-guided implant surgery as easy, safe, and predictable [16,29]. A recent clinical study on implants placed using flapless-guided surgery and immediate loading reported successful results [16]. However, we must bear in mind that it is not a technique free of problems, due to the difficulty of having visual control of the tissues [30]; it can even, on rare occasions, cause severe complications such as those described by Limongelli et al. in a clinical case [31]. This time, 110 implants were installed (65 implants in fresh sockets) in 12 patients with a guided surgery system. All implants were immediately loaded by means of fixed provisional full-arch restorations and followed for a period of one year. The outcome variables were implant stability at placement, implant survival, complications, prosthesis success, soft tissue stability, and patient satisfaction. After 6 months of provisionalization, 72 fixed prosthetic restorations (53 single crowns, 17 bridges, and 2 fixed full arches) were delivered. At the end of the study, a high implant survival rate (98.2%) was reported, with only 2 implants that had failed. The study concluded that flapless-guided implant surgery is a reliable and successful procedure, capable of guaranteeing adequate soft tissues and showing favorable aesthetic outcomes [16].

Another clinical study, with a 7-year follow-up, evaluated the cumulative survival rate of dental implants placed using computer tomography (CT) guided surgery including CBCT [29]. Virtual planning was performed using guided implant software. Stereolithographic guides were used to place fully guided implants according to the planned depths and angulations. In total, 796 implants were placed in 177 patients. Of that group, 43 patients were restored with full-arch reconstructions from a total of 314 implants placed. Additionally, 34 patients received implants in the maxilla and 9 patients in the mandible. Of the 314 implants placed, there were 8 failures (2.5%). A total of 145 implants were immediately loaded with provisional restorations (full-arch restoration and All-on-4/5/6), reporting 3 failures (2.1%). Clinical findings from this study strongly suggest that the implant treatment with guided surgery and immediate loading with full-arch restorations demonstrated higher survival rates and similar long-term outcomes when compared with conventional implant placement [29].

Guided implant surgery increases the ability to insert implants more precisely, especially in fully edentulous cases, with an important reduction of surgery duration, better clinical conditions after surgery, and the possibility of placing a provisional restoration for immediate loading [11,12,13,14,15,16]. In the present study, 22 patients received 198 implants, inserted through a flapless-guided surgery and immediate loading with provisional fixed full-arch restorations. After a provisionalization period of 6 months, 22 definitive fixed full-arch restorations were delivered. Only 5 implants failed, with a 7-year implant survival rate of 97.5%.

Despite high survival rates for implants placed using computer-guided surgery, an important rate of prosthetic and biologic complications has also been reported [31,32,33,34,35]. In a systematic review, complications related to implants placed using guided surgery for the treatment of fully edentulous patients were evaluated [31]. Low primary implant stability was the most common surgical complication in the different studies. Implant loss (2.5%) was predominantly related with a failure in osseointegration (early losses). Mucositis was the most frequent biological complication and was related to poor hygiene. Peri-implantitis (13.7%) was also reported and associated with implant loss. The most frequent prosthetic complication was fracture, which occurred in both provisional and definitive prostheses. Screw loss or loosening, loss of implant and abutment fit, or loss of abutment and prosthesis fit were also frequent [31].

Prosthodontic complications were very frequent in the present study. Six patients (27.3%) showed technical problems with restorations (resin fracture, ceramic chipping, loss/fracture of prosthetic screw). Technical complications are relatively frequent in studies of patients treated with guided surgery and immediate loading [16]. A retrospective study reported complications in 33.4% of patients treated with complete-arch fixed reconstruction by means of guided surgery and immediate loading with 1 year of follow-up [16]. Similar clinical outcomes are reported in another study of immediate loading implants installed in edentulous jaws following computer-assisted treatment planning in 29 edentulous patients [32]. A total of 176 fixtures were installed to support 21 maxillary and 10 mandibular reconstructions. Patients were followed for up to 44 months. Implant-supported suprastructures remained stable during the follow-up period in 26 out of 31 jaws (90% maxilla, 70% mandible). Technical complications occurred in 42% of treated cases. Misfitting of abutment bridges appeared in five cases, resulting in disconnection of the bridge in two patients where fixtures were left for unloaded healing. Extensive adjustments of occlusion were made in 10% of the immediately connected bridges [33].

Smoking is an important risk factor for implant survival rate. The results of the present report suggest that smoking alters host immune response of peri-implant tissues and increases the susceptibility for biological implant complications in guided surgery [35]. During the follow-up control, biologic complications (i.e., peri-implantitis) were reported (9.3% of implants). In fact, the prevalence of implant failures (50%) and peri-implantitis (66.6%) were significantly more frequent in smoking patients. Moreover, in patients with smoking habits, the marginal bone loss was significantly higher. These results are confirmed in a 5-year clinical study about implants inserted in 30 completely edentulous patients using a flapless-guided surgery and immediate loading with fixed complete dentures [34]. Nine (4.9%) implants failed. Of the 9 failures, 8 occurred in 3 smoking patients. The survival rate for all patients was 91.5% (81.2% in smoking patients and 98.9% in non-smoking patients). The mean marginal bone resorption was 2.6 mm and 1.2 mm in smoking and non-smoking patients [34].

## 5. Conclusions

The antecedents of peri-implantitis are critical elements for the survival of the implants. The loss of implants was significant in patients who smoked up to 10 cigarettes compared to non-smokers. Peri-implantitis is one of the key elements in the long-term follow-up of implants, and it was more manifest in smoking patients and in those with a history of peri-implantitis. Marginal bone loss was more significant in smokers. Full-arch rehabilitation is presented as a predictable alternative with minor fatigue problems that are easily solvable.

## Figures and Tables

**Figure 1 ijerph-18-04125-f001:**
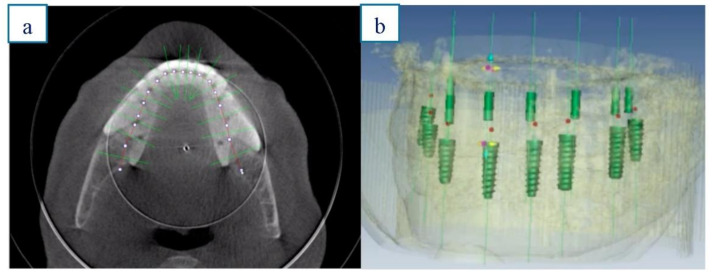
a & b.

**Figure 2 ijerph-18-04125-f002:**
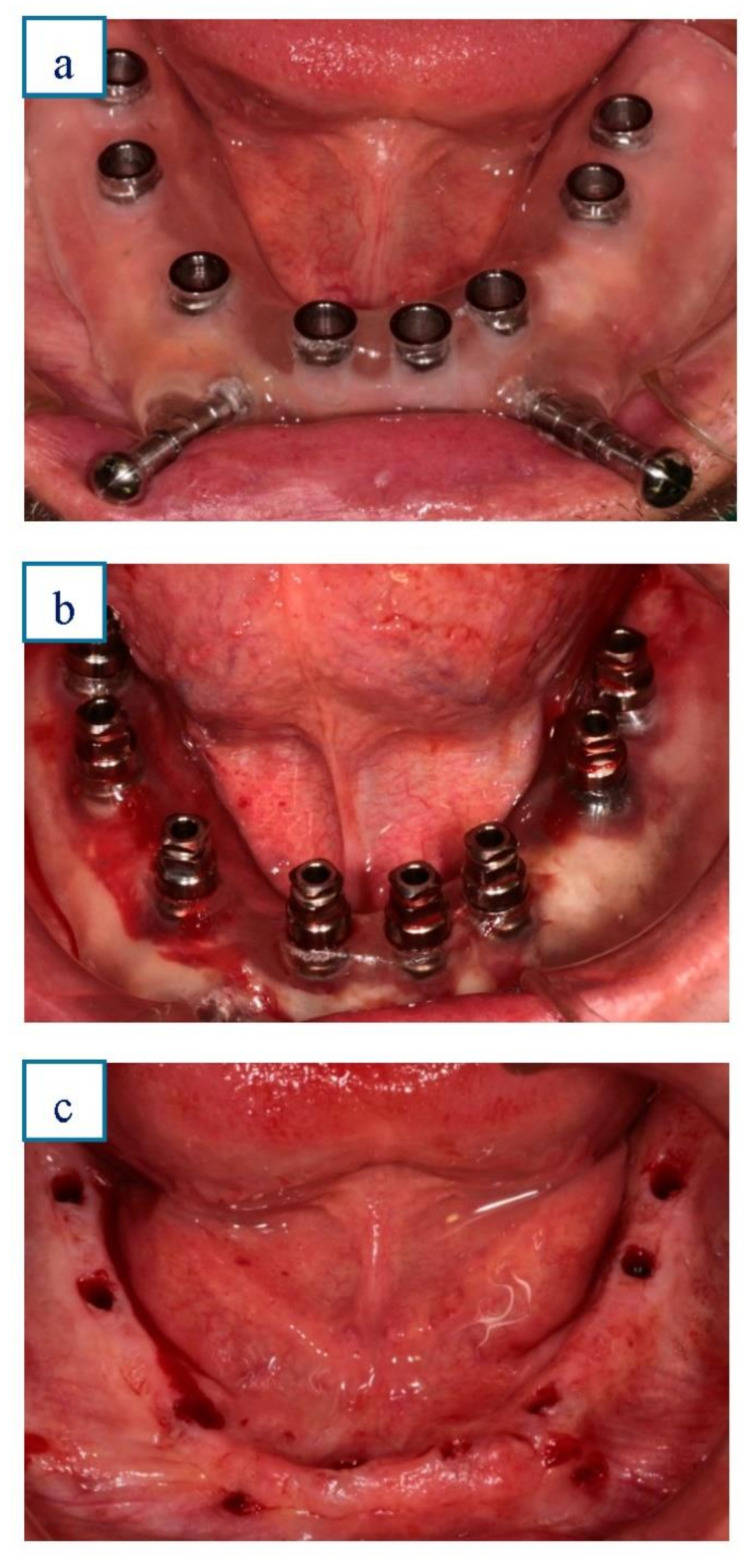
a, b, & c.

**Figure 3 ijerph-18-04125-f003:**
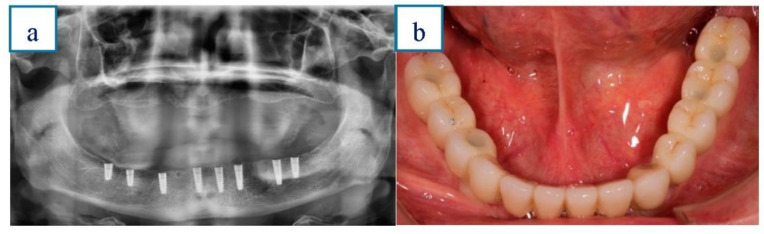
a & b.

**Table 1 ijerph-18-04125-t001:** Description of the main features of the studied population.

	*n*	%
Males	10	45.4
Females	12	54.5
History of periodontitis	11	50
Smokers	6	27.3
History of periodontitis and smokers	5	22.7

*n* = patient.

**Table 2 ijerph-18-04125-t002:** Description of the width and length of the implants placed and the percentage of implants lost. The 5 lost implants are distributed: 2 of 3.5 mm diameter and 3 of 4 mm, with *p* = 0.629 and 4 of 10 mm in length and 1 of 12, with *p* = 0.703.

	*n*	%
3.5 mm implant diameter	87	43.9
4 mm implant diameter	111	56.1
10 mm implant length	132	66.7
12 mm implant length	66	33.3
Implant loss	5	2.5

*n* = implant.

**Table 3 ijerph-18-04125-t003:** Description of patients with complications.

		*n*	%	*n* Total	% Total
Implant loss		5	22.7	5	22.7
Peri-implantitis	History of periodontitis	7	63.6	10	45.4
	Smoking	4	66.6 *		
Technical complications	Provisional prosthesis	2	9.1	6	27.3
	Definitive prosthesis	4	18.2		

* *p* < 0.05 *n* = patient.

## Data Availability

The associated data of a statistical nature, can be requested to: jl.lopez@ub.edu & evelasco@us.es

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
