# Peer review of "Immediate Loading of Implants Placed by Guided Surgery in Geriatric Edentulous Mandible Patients"

_ijerph, 2021, doi:10.3390/ijerph18084125_

Round 1

Reviewer 1 Report

Abstract: From abstract section delete the terms: introduction, materials and methods result and conclusion. 

Materials and methods: please add the number and protocol of the ethical committee approval;

Result:

1)Patients enrolled in this study are all healthy patients? any type of drug was administered? 

2) all the patients defined as smokers in this section smoke less than 10 cigarettes? 

3) why have you inserted such high number of implants per each patients? (average 9) 

4) have you distinguished survival rate considering the diameter and length? 

Discussion: in this section the authors should insert surgical complications of flapless technique even rare such as some described in this article

Limongelli L, Tempesta A, Crincoli V, Favia G. Massive Lingual and Sublingual Haematoma following Postextractive Flapless Implant Placement in the Anterior Mandible. Case Rep Dent. 2015;2015:839098. doi: 10.1155/2015/839098. Epub 2015 May 17. PMID: 26075110; PMCID: PMC4449918.

Conclusion section must be improved. 

Author Response

Review 1

#Abstract: From abstract section delete the terms: introduction, materials and methods result and conclusion. 

Response:

Done, the new abstract is (page 1, line 18-38)

“The aim of this study was to show the clinical outcomes of immediate loading of implants inserted by guided surgery in edentulous mandibular patients.

Mandibular edentulous patients were diagnosed with oral examination, cone beam computerized tomography and diagnostic casts for intermaxillary relations and treated with 8-10 implants for rehabilitation with guided surgery and immediate loading. After flapless surgery, implants were loaded with an immediate acrylic temporary prosthesis. After a period of six months, a ceramic definitive full arch prosthesis was placed.

Twenty-two patients (12 females, and 10 males) were treated with 198 implants. Eleven patients (50%) had a previous history of periodontitis. Six patients (27.3%) were smokers. The follow-up was 84.2 ± 4.9 months. Clinical outcomes showed a global success of 97.5% of implants. Five implants were lost during the healing phase with provisional prosthesis. Twenty-two fixed full-arch rehabilitations were placed in the patients over the 193 remaining implants. Mean marginal bone loss was 1.44 mm ± 0.45 mm. Six patients (27.3%) showed some kind of mechanical prosthodontic complications. Eighteen implants (9.3%) of the 193 remaining implants were associated with peri-implantitis.

This study indicates that treatment of mandibular edentulous geriatric patients with full-arch fixed rehabilitations by guided surgery and immediate loading of implants placed appears to be a successful implant protocol.

#Materials and methods: please add the number and protocol of the ethical committee approval;

Response:

This is a retrospective study carried out in a teaching center. In it, patients sign a consent so that their data (anomimized) can be used for studies that revert to teaching and research. In our case, it is part of a project started years ago with the corresponding endorsement of the ethics committee of the University of Seville on July 11, 2013. Supporting PDF document.

#Result:

1)Patients enrolled in this study are all healthy patients? any type of drug was administered? 

Response:

Thank you very much for your comment, we have added a sentence to clarify the issue. (Page 2, lines 89-91):

Thus, the patients included in this study are ASA I-II patients, without decompensated systemic diseases or medication that may interfere with the osseointegration of the implants.

#2) all the patients defined as smokers in this section smoke less than 10 cigarettes? 

Response:

Yes, as we indicated in the exclusion criteria, if more than 10 cigarettes were smoked, they were excluded. (Page 2, line 88)

#3) why have you inserted such high number of implants per each patients? (average 9) 

Response:

Thank you very much for your comments A high percentage of patients have a history of periodontitis (50%) and 22.7% have a history of periodontal disease and are also smokers. If also, we take into account that 43.9% of the implants are 3.5 mm in diameter. We found it appropriate to place between 8 and 10 impants, whenever possible in the pcaientes to improve the loads. The mean follow-up was 84.2 months, 7 years. And a few years ago it was frequent to resort to a greater number of implants for complete rehablitations. In fact, there was frequent consensus among the authors [among others Goiato et al, Hobikirk ete al, Rosentiel et al, Stevenson et al, May & Romnos] that a greater number of implants for a given prosthetic space would better withstand chewing loads and dissipate stress in the bone more effectively. In these omenteos it is possible that we had resorted to placing between 6 and 8 and making shorter plows:

-Goiato MC, Santiago Junior JF, Pellizzer EP, Moreno A, Villa LM, Dekon SF, de Carvalho PS, dos Santos DM. Systemic Trans- and Postoperative Evaluations of Patients Undergoing Dental Implant Surgery. Clinics (Sao Paulo). 2016 Mar;71(3):156-62.

-Hobkirk JA, Wiskott HWA. Biomechanical aspects of oral implants.Consensus report of working group I.Clin Oral Implant. Res.2006; 17 (suppl.2): 52-4.

-Rosentiel S, Land M, Crispin B. Dental Luting agents: A review of the current literature. J Prosthet Dent. 2004; 80: 280-301.

-Stevenson W, Harrod JT, Van Eyck MN. Retrospective analysis of 56 edentulous dental arches restored with 344 single-stage implants using an immediate loading fixed provisional protocol: statistical predictors of implant failure. Int J Oral Maxillofac Implants. September 1, 2007; 22(5): 823-30.

-May D, Romanos GE. Immediate implant-supported mandibular overdentures retained by conical crowns: A new treatment concept. Quintessence Int. 2006; 33: 5-12.

#4) have you distinguished survival rate considering the diameter and length? 

Response:

Thank you very much for the suggestion, the data referring to the diameter and length of the failed implants are not significant and have been added in the legend of Table 2. (Page 6, lines 183-185).

#Discussion: in this section the authors should insert surgical complications of flapless technique even rare such as some described in this article

Limongelli L, Tempesta A, Crincoli V, Favia G. Massive Lingual and Sublingual Haematoma following Postextractive Flapless Implant Placement in the Anterior Mandible. Case Rep Dent. 2015;2015:839098. doi: 10.1155/2015/839098. Epub 2015 May 17. PMID: 26075110; PMCID: PMC4449918.

Response:

Thank you very much for your comment we have added a paragraph and two new references. (Page 7, lines 241-243)

However, we must bear in mind that it is not a technique free of problems, derived above all from the difficulty to have a visual control of the tissues [30]; it can even on rare occasions cause severe complications such as that described by Limongelli et al in a clinical case [31].

#Conclusion section must be improved. 

Response:

Thank you for your comments, we have modified the conclusions (Page 1, lines 33-38; and page 9, lines 310-315)

The antecedents of perimplantiis are a critical element for the survival of the impantes. The loss of implants was significant in patients who smoked up to 10 cigarettes compared to non-smokers. Preimpantitis is one of the key elements in the long-term follow-up of implants and it was more manifest in smoking patients and in those with a history of peri-implantitis. Marginal bone loss was more significant in smokers. Full arch rehab is presented as a predictable alternative with minor fatigue problems that are easily solvable.

Reviewer 2 Report

Dear authors,  

please change with the correct punctuation inside the square brackets at lines 63,64; 65; 204. 

Please place the phrase at the line 93,94 in the correct context ( When you described the surgical phases). 

Please explain at the line 123 the acronym  ISQ 

You write that " The cumulative survival rate...was 97.5%. ", but also that 10 patients had a peri-implantitis in 18 implants. It is necessary to specify what degree of mobility, if they have been treated and if they actively participate in chewing. This figure influences the implant success. Please specify these data better.

Specify why it was chosen to place 8 implants in edentulous mandibles and not a smaller number. Argue with the risks and benefits of the protocol.

The captions of the tables are too concise, we need to reformulate them better. 

Author Response

#Dear authors,  

please change with the correct punctuation inside the square brackets at lines 63,64; 65; 204. 

Response:

Thank you very much for your correction Done in red

#Please place the phrase at the line 93,94 in the correct context ( When you described the surgical phases). 

Response:

Thank you very much for your correction.

We have changed the position phrase, in red (Page 3, line 97-98)

... All patients were treated under local anesthesia using articaine with adrenaline.

#Please explain at the line 123 the acronym  ISQ 

Response:

Thank you very much for your correction. Done, in red (Page 4, line 129)

[resonance frequency analysis]

#You write that " The cumulative survival rate...was 97.5%. ", but also that 10 patients had a peri-implantitis in 18 implants. It is necessary to specify what degree of mobility, if they have been treated and if they actively participate in chewing. This figure influences the implant success. Please specify these data better.

Response:

Indeed, in 18 implants peri-implantitis was detected, but without any degree of associated mobility. When the implant moved, it entered the group of failed implants.

#Specify why it was chosen to place 8 implants in edentulous mandibles and not a smaller number. Argue with the risks and benefits of the protocol.

Response:

Thank you very much for your comments A high percentage of patients have a history of periodontitis (50%) and 22.7% have a history of periodontal disease and are also smokers. If also, we take into account that 43.9% of the implants are 3.5 mm in diameter. We found it appropriate to place between 8 and 10 impants, whenever possible in the pcaientes to improve the loads. The mean follow-up was 84.2 months, 7 years. And a few years ago it was frequent to resort to a greater number of implants for complete rehablitations. In fact, there was frequent consensus among the authors [among others Goiato et al, Hobikirk ete al, Rosentiel et al, Stevenson et al, May & Romnos] that a greater number of implants for a given prosthetic space would better withstand chewing loads and dissipate stress in the bone more effectively. In these omenteos it is possible that we had resorted to placing between 6 and 8 and making shorter plows:

-Goiato MC, Santiago Junior JF, Pellizzer EP, Moreno A, Villa LM, Dekon SF, de Carvalho PS, dos Santos DM. Systemic Trans- and Postoperative Evaluations of Patients Undergoing Dental Implant Surgery. Clinics (Sao Paulo). 2016 Mar;71(3):156-62.

-Hobkirk JA, Wiskott HWA. Biomechanical aspects of oral implants.Consensus report of working group I.Clin Oral Implant. Res.2006; 17 (suppl.2): 52-4.

-Rosentiel S, Land M, Crispin B. Dental Luting agents: A review of the current literature. J Prosthet Dent. 2004; 80: 280-301.

-Stevenson W, Harrod JT, Van Eyck MN. Retrospective analysis of 56 edentulous dental arches restored with 344 single-stage implants using an immediate loading fixed provisional protocol: statistical predictors of implant failure. Int J Oral Maxillofac Implants. September 1, 2007; 22(5): 823-30.

-May D, Romanos GE. Immediate implant-supported mandibular overdentures retained by conical crowns: A new treatment concept. Quintessence Int. 2006; 33: 5-12.

#The captions of the tables are too concise, we need to reformulate them better. 

Response:

Thank you very much for your correction, we have changed the legend of Tables 1 and 2. (Page 5, line 169; Page 6, line 183 &185)

Table 1. Description of the main features of the studied population.

Table 2. Description of the width and length of the implants placed and the percentage of implants lost. The 5 lost implants are distributed: 2 of 3.5mm diameter and 3 of 4mm, with p = 0.629 and 4 of 10 mm in length and 1 of 12, with p = 0.703.

Round 2

Reviewer 1 Report

This reviewer appreciated the changes applied. Now the article is suitable for publication 

Reviewer 2 Report

Dear authors, 

now is very better.

Please insert your comment  " in 18 implants peri-implantitis was detected, but without any degree of associated mobility. When the implant moved, it entered the group of failed implants" in the paper. It is right to specify for readers otherwise is not clear. 

Thank you 

Kind Regards